# Evaluation of community pharmacists' knowledge and attitude about Hajj and Umrah-related health conditions in the western region, Saudi Arabia: A cross-sectional study

**Nasser M. Alorfi**[ID]*, **Ahmed M. Ashour, Maan H. Harbi, Fahad S. Alshehri**[ID]

Department of Pharmacology and Toxicology, College of Pharmacy, Umm Al-Qura University, Makkah, Saudi Arabia

* nmorfi@uqu.edu.sa

## Abstract

### Background

During the Hajj and Umrah seasons in Saudi Arabia, pilgrims tend to experience a higher frequency of various health conditions. Respiratory infections, gastrointestinal infections, and food poisoning are among the most prevalent ailments. To address these health concerns, community pharmacists (CPs) have developed standardized treatment protocols. Pharmacists' skills in medication dispensing, health consultations, and preventive care enhance pilgrims' well-being in challenging pilgrimage settings.

### Aims

The objective of this study was to investigate the knowledge and attitude of CPs towards health conditions related to Hajj and Umrah in the Western Region of Saudi Arabia where the Hajj and Umrah are taking place.

### Methods

Between March and April 2023, an online cross-sectional study using Google form was carried out among CPs who work in the western region of Saudi Arabia. The study made use of a self-administered questionnaire consisting of four sections that aimed to obtain information about the CPs' knowledge and attitudes towards health conditions related to Hajj and Umrah. Data analysis was conducted using the Statistical Package for the Social Sciences version 26 (SPSS).

### Result

A total of 496 CPs completed and returned the questionnaire, by giving a response rate of 99.2% (n = 500). Among them, 55.1% were aware of the necessary vaccines for Hajj and Umrah. Approximately 46.6% of CPs provided health-related advice to pilgrims. The most

**Data availability statement:** The relevant data can be accessed with the following DOI: (https://doi.org/10.6084/m9.figshare.28357232).

**Funding:** The author(s) received no specific funding for this work.

**Competing interests:** The authors have declared that no competing interests exist.

**Abbreviation:** CPs, Community pharmacists; LF, Lymphatic filariasis (LF); MERS, Middle East Respiratory Syndrome; SCFHS, Saudi Commission for Health Specialties; SPSS, Statistical Package for the Social Sciences.

common health conditions experienced by pilgrims were diarrhea (59.5%), followed by flu and cough (58%), gastrointestinal diseases (39.4%), food poisoning (33.6%), viral fever, and heat stroke (24.4%). Regarding attitudes towards vaccination, 55.3% of CPs agreed that vaccination is safe for pilgrims aged 65 years and above, and 65.7% agreed that vaccination can help reduce medical costs during Hajj and Umrah seasons. Additionally, 61.1% of CPs recommended updating immunization against vaccine-preventable diseases for all travelers to ensure a safe Hajj and Umrah. The overall mean knowledge of CPs regarding vaccination during Hajj and Umrah was 4.739(2.49) (median = 5; Range = 0-9). However, 56.7% of CPs demonstrated good knowledge, while 43.3% reported poor knowledge regarding vaccination during Hajj and Umrah.

## Conclusion

The community pharmacists (CPs) in the Mecca region were found to have good knowledge, with more than half of them having positive attitudes about vaccination for Hajj and Umrah-related health conditions. It is crucial to enhance the knowledge and attitudes of CPs to provide better care and participate in reliable and supportive healthcare and counseling sessions for managing various health infections.

## Introduction

The Hajj and Umrah are the pillars of Islam that attract people from all around the world [1,2]. The Hajj is an annual pilgrimage that happens in the city of Makkah, Saudi Arabia. It is regarded as a religious requirement for all adult Muslims worldwide who have the physical and financial ability, and an estimated 2-3 million peoples attend each year [1–3]. While the Hajj can be performed in the dhul Hajj month, once a year, on the other hand, Umrah can perform at any time of the year [1,2]. The peak period for Umrah is when pilgrims travel to Mecca to undertake Umrah during Ramadan. During gatherings or crowded places, there were always higher chances of the development of various infections [4,5].

According to the literature, a diverse range of nations were represented by the majority of pilgrims, particularly those performing Hajj, such as Indonesia, Pakistan, India, Malaysia, Nigeria, and Egypt [6]. Dengue fever, lymphatic filariasis (LF), soil-transmitted helminth diseases, and leprosy were more widespread in tropical nations in South and Southeast Asia [1,2,4,5,7]. Although previous research has shown that respiratory illnesses are the most common among pilgrims [4,5] with a prevalence of up to 90% [8]. Another recent comprehensive analysis, on the other hand, revealed a rhinovirus prevalence of 5.9-48.8%, followed by influenza virus at 13.9% and non-Middle East Respiratory Syndrome coronavirus at (MERS) coronaviruses at 13.2% [9] Similarly, Salmon-Rousseau et al [4,5] reported respiratory tract infections, ENT infections, pyogenic pneumonia, whooping cough, TB, and meningococcal meningitis. Among pilgrims, additional waterborne diseases like gastroenteritis and hepatitis A were widespread [4,5].

It was evidenced that respiratory infections are the most common infections in the holy cities and the majority of visitors who come from different nations to perform their rituals develop at least one form of respiratory infection or another during their stay in holy cities including Madina [9–12]. The incidence of illnesses during the mass gatherings can be attributed to several factors, including overcrowding, changes in climatic conditions, pollution, and shared accommodation. Additionally, it is worth considering that many pilgrims may be elderly individuals aged 50 years or older with chronic diseases and comorbidities, such as hypertension and diabetes [13,14].

Pharmacists, as front-line health care professionals must recognize possible infections, during the Hajj and Umrah to reduce the risk of health-related complications, which can result in chronic conditions, comorbidities, increased costs to the patients and may lead to endemics or pandemics, if not treated or recognized promptly [15–18]. As a result, health care providers in all practice settings must be vigilant in checking for potential risks of infections and diseases and, prescribe medications for the patients to control the infectious diseases as well as patient counseling on foods and other beverages [16–18]. For example, investigations have discovered that mass gathering is more likely to cause various infections including food, allergic reactions, flu, and cough [10,13,19,20]. Furthermore, multinational pilgrims when combined, the possibilities of infections can be further higher, which may increase the risk of unwanted consequences [19,20].

Pharmacists play an important role in the individual's, health and management from whom people seek medical guidance [17,18]. In this view, CPs were playing a significant role in controlling various infections, therefore posing adequate knowledge helps CPs to be more vigilant about the recognition of new strains of infections at their practice site. Their expertise in dispensing medications, offering health consultations, and promoting preventive care contributes to the overall health and safety of pilgrims in the challenging and physically demanding pilgrimage environments. The accessibility and guidance provided by community pharmacists contribute significantly to the pilgrims' ability to manage health conditions effectively, fostering a safer and more fulfilling religious experience during Hajj and Umrah.

There have been several reports published previously from around the world to study the knowledge, attitudes, and practice of pharmacists towards the prevention of infections in the Hajj and Umrah. As per our knowledge, no such research has been conducted so far among Saudi pharmacists working in the western region of Saudi Arabia. Hence, the present study was designed to investigate knowledge and attitude about Hajj and Umrah-related health conditions in the Western Region, of Saudi Arabia.

The rationale for the study stems from the significance of the annual Hajj and Umrah pilgrimages in the region and the pivotal role community pharmacists play in public health. Given the large influx of pilgrims during these religious events, there is a heightened risk of various health conditions.

## Method

### 2.1. Study design

A cross-sectional, online survey study was conducted between 1st March to 30th April 2023 among Community pharmacists (CPs) living Makkah, Madinah, Jeddah and Taif which represent the western region of Saudi Arabia to evaluate their knowledge and attitude about Hajj and Umrah-related health conditions.

### 2.2. Survey respondent selection process and data collection

It is a prospective self-reporting survey using online questionnaire involving CPs who are registered with the Saudi Commission for Health Specialties (SCFHS), currently practicing in the community pharmacies, and CPs who agreed to provide informed consent were included in the study. The study excluded CPs from the other regions of Saudi Arabia and those who are not in the practice. The study was conducted according to the guidelines of the Declaration of Helsinki and approved by the research ethical committee at Umm Al-Qura University, Makkah, Saudi Arabia (approval number. HAPO-02-K-012-2022-11-1314). Before data collection the CPs were informed about the objectives of the study and verbal informed consent was obtained from all respondents emphasizing the confidentiality and use of personal data for the

research work only. Furthermore, participation in this study was voluntary, no monetary or non-monetary rewards were provided, and participants were free to leave at any time.

## 2.3. Sample size estimation

A simple sampling method was utilized, and the required sample size was calculated using an online sample size calculator (https://www.openepi.com/SampleSize/SSCohort.htm), with a 95% confidence level and 5% margin of error. Since the outcomes for each question were unknown, a response distribution of 50% was assumed [21]. Based on this calculation, the sample size was determined to be 384 subjects. However, to enhance the data's reliability, we aimed to survey around 500 pharmacists.

## 2.4. Questionnaire design

The questionnaire employed in this study was formulated by conducting a thorough review of the available literature on health-related conditions during Hajj and Umrah, as well as the official websites of the Saudi Ministry of Health [19,20]. The questionnaire for this study was structured into four distinct sections, encompassing a total of 32 questions. The first section consisted of nine questions aimed at gathering demographic information about the participants, such as age, gender, education level, year of experience, country of qualifications, and pharmacy location. Additionally, this section inquired about the advice, services, and time devoted to Hajj and Umrah health-related conditions. The second section comprised 16 questions designed to measure the participants' knowledge of Hajj and Umrah health-related conditions. The questions were presented in multiple-choice format, with some questions using a 3-point scale (Yes/No/I don't know) to elicit responses. The third section collected information on the attitudes of CPs towards vaccination and related health conditions among pilgrims. Participants were asked to respond on a 3-point scale (Yes/No/I don't know) regarding which conditions to seek health behavior for. Lastly, the fourth section focused on the medication used to manage influenza and other related conditions.

The questionnaire developed for this study underwent a two-step validation process. First, the initial draft was reviewed by a group of research experts, and secondly, a pilot study was conducted among a small sample of 15 individuals to ensure readability and ease of administration. Feedback from the pilot study was incorporated into the final questionnaire, which had a reliability coefficient of knowledge items 0.85, attitudes 0.72 and awareness 0.74 as measured by Cronbach's Alpha, indicating its suitability for the study. The final version was distributed to CPs in the selected locations using an online survey tool, with one reminder sent to encourage participation. The pilot study data was not included in the final analysis.

To assess the knowledge of CPs about vaccination, a total of 10 knowledge items were used. A score of 'one' was given for each correct answer, and a score of 'zero' for each incorrect answer. The overall knowledge score was determined by adding up the scores for all 10 items. The knowledge score was then categorized as 'Good' for scores greater than 50% of the total knowledge score, and 'Poor' for scores less than 50% of the total knowledge score[22,23].

## 2.5. Statistical analysis

The collected data was analyzed using Statistical Package for Social Sciences version 26.0 (SPSS Inc., Chicago, IL, USA). Descriptive statistics such as frequencies (*n*) and percentages (*%*) were computed. The knowledge scores were tabulated for the knowledge items. To investigate the relationship between the demographic characteristics of the CPs and their knowledge, the multiple linear regression was employed. A significance level of $p < 0.05$ was set to determine statistical significance.

## Results

The survey was completed by a total of 496 CPs by giving a response rate of 99.2% (n = 500). The majority of the respondents (71.8%) were male, while 28.2% were female. Among the CPs, approximately 37% (n = 182) were aged between 31 and 35, and 27.2% were aged between 26 and 30. In terms of experience, 35.1% had 6-10 years of experience, and 28.8% had 11-15 years of experience. A majority of the CPs (55.4%) held a bachelor's degree in pharmacy, while 27.2% held a PharmD degree. Regarding their country of qualification, 45% of the CPs were qualified in Egypt, 18.5% in Saudi Arabia, and 13.5% in Yemen. More than half of the respondents (53.2%) were urban dwellers. Table 1 provides detailed information about the demographic characteristics of the CPs.

The results of the study indicate that a majority of CPs, 55.1%, had knowledge of the required vaccines for Hajj and Umrah. When it came to providing advice or services related to Hajj and Umrah health-related diseases, 46.6% of CPs reported offering health advice to pilgrims while 32.1% did not. Interestingly, 46.6% of CPs also reported receiving or seeing a pilgrim at the pharmacy complaining of infection. A small proportion of CPs, 14.5%, reported

**Table 1. Demographic characters of the community pharmacist (n = 496).**

| Variables | Frequencies (n) | Percentage (%) |
|---|---|---|
| **Gender** | | |
| Male | 356 | 71.8% |
| Female | 140 | 28.2% |
| **Age(Years)** | | |
| 21-25 | 32 | 6.5% |
| 26-30 | 135 | 27.2% |
| 31-35 | 182 | 36.7% |
| 36-40 | 95 | 19.2 |
| >41 | 52 | 10.5% |
| **Years of experience** | | |
| 0-5 years | 103 | 20.8% |
| 6-10 years | 174 | 35.1% |
| 11-15 years | 143 | 28.8% |
| >15 years | 76 | 15.3% |
| **Educational level** | | |
| Diploma in pharmacy\Technician | 34 | 6.9% |
| Bachelor\Bpharm | 275 | 55.4% |
| PharmD (Doctor of Pharmacy) | 135 | 27.2% |
| Postgraduate | 52 | 10.5% |
| **Country of qualification** | | |
| Saudi Arabia | 92 | 18.5% |
| Egypt | 223 | 45% |
| Yemen | 68 | 13.7% |
| Pakistan | 58 | 11.7% |
| India | 32 | 6.5% |
| Other countries | 23 | 4.6% |
| **Pharmacy location** | | |
| Rural | 232 | 46.8% |
| Urban | 264 | 53.2% |

spending more than 1 hour per day offering Hajj and Umrah health-related advice to pilgrims, while 14.3% reported spending less than 1 hour per day. More detailed responses regarding the characteristics of the services provided by CPs can be found in Table 2.

The results of this study showed that diarrhea (59.5%, n = 234) was the most common health condition reported during Hajj and Umrah seasons, followed by flu and cough (58%, n = 228), gastrointestinal diseases (39.4%, n = 155), food poisoning (33.6%, n = 132), and viral fever and heat stroke (24.4% and 24.4%, respectively; n = 96 each). A breakdown of the responses provided by CPs regarding the most prevalent infectious diseases during these seasons can be found in Fig 1.

Table 3 outlines the extent of knowledge among CPs regarding diarrhea and its management. The results indicated that a significant proportion of CPs (n = 143, 37.8%) identified Enter Toxigenic E. coli as the most common causative agent of diarrhea, followed by Salmonella (21.2%, n = 80) and Rotaviruses (16.9%, n = 64) (as presented in Table 2). In terms of managing diarrhea, 48.3% of CPs (n = 185) believed that no active treatment is necessary, and only maintaining hydration using Gastrolyte® and allowing diarrhea to resolve naturally. In contrast, 27.7% of CPs (n = 106) suggested starting treatment immediately with loperamide while consuming plenty of fluids. A smaller proportion of CPs (n = 21, 5.5%) recommended one dose of doxycycline 200 mg in addition to loperamide and fluid intake to treat diarrhea. Further details of the CPs' responses are provided in Table 3.

## 3.1. Community pharmacist responses towards First aid items recommendations to pilgrims

The responses of CPs regarding the most recommended first aid items for pilgrims were analyzed. More than half of the CPs (53.6%; n = 266) suggested oral rehydration sachets, followed by face masks recommended by 39.7% of the CPs (n = 197), and painkillers or paracetamol tablets by 38.7% of the CPs (n = 192). Additionally, 37.95% of the CPs recommended using

**Table 2. CPs' responses towards awareness of the required vaccines for Hajj and Umrah and providing health-related advice to the pilgrims.**

| Variables | Frequency (n) | Percentage (%) |
|---|---|---|
| **Do you aware of the required vaccines for Hajj and Umrah Health** | | |
| Yes | 212 | 55.1% |
| No | 75 | 19.5% |
| I don't know | 98 | 25.5% |
| **Do you provide Hajj and Umrah health-related conditions advice or services to the patients?** | | |
| Yes | 231 | 46.6% |
| No | 159 | 32.1% |
| I don't know | 106 | 21.4% |
| **Have You Seen or received a pilgrim to the pharmacy about Complaining of infection/ problem?** | | |
| Yes | 231 | 46.6% |
| No | 198 | 39.9% |
| **How much time or days do you spend on providing Hajj and Umrah health-related conditions advice to pilgrims?** | | |
| Less than 1 hour per day | 71 | 14.3% |
| More than 1 hour per day | 72 | 14.5% |
| Less than 1 hour per week | 78 | 15.7% |
| More than 1 hour per week | 50 | 10.1% |
| Everyday | 95 | 19.2% |
| Less than 1 hour in a month | 53 | 10.7% |

sunscreen or hats to protect against the sun, while 35.3% suggested carrying a small range of bandages or dressings (n = 175). The detailed responses of the CPs regarding the recommended first aid kits for pilgrims were provided in Fig 2.

In addition, a considerable proportion of CPs (38.5%; n = 191) believed that influenza vaccination would be administered to pilgrims before their departure to Saudi Arabia. This was followed by Meningococcal Meningitis at 14.1% (n = 70), polio at 11.1% (n = 55), Hepatitis A at 8.9% (n = 44), Tetanus at 8.3% (n = 41), and Measles at 7.5% (n = 37). For a comprehensive overview of the CPs' responses on the most commonly known vaccines given before traveling to Saudi Arabia, refer to Fig 3.

Despite the fact that 57.1% of the CPs acknowledged the necessity of vaccination to safeguard and impede the spread of infectious diseases before visiting Saudi Arabia, 64.8% of

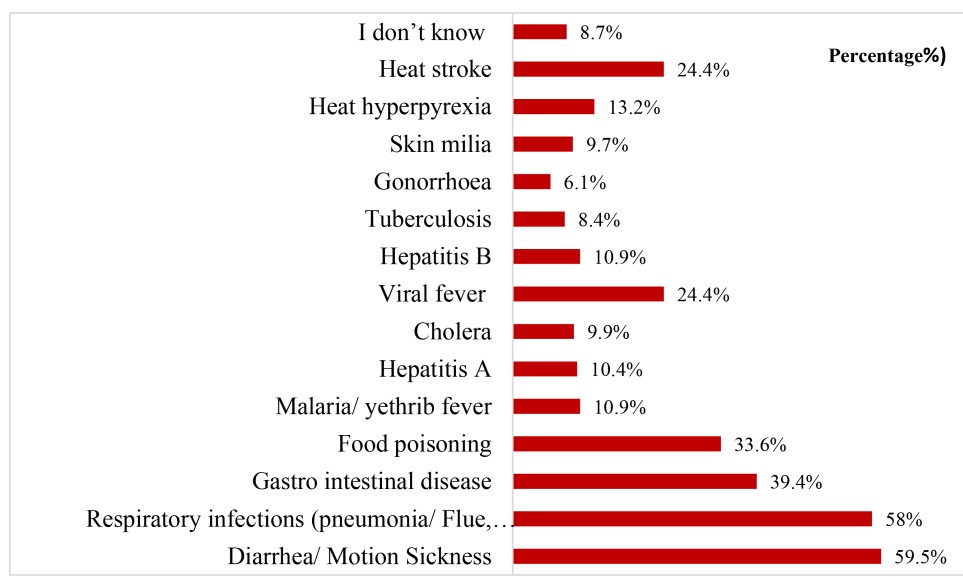

**Fig 1. CPs responses towards the most common Hajj and Umrah health-related infectious diseases.**

**Table 3. Knowledge of CPs towards diarrhea and its management.**

| Characters | Frequency (n) | Percentage (%) |
|---|---|---|
| **The most common causative organism for diarrhea during Hajj and Umrah** | | |
| Enter toxigenic E. coli (ETEC) | 143 | 37.8% |
| Giardia intestinalis | 57 | 15.1% |
| Salmonella | 80 | 21.2% |
| Rotaviruses | 64 | 16.9% |
| Campylobacter pylori | 34 | 9% |
| **The preferred method for treatment option of diarrhea during Hajj and Umrah** | | |
| No active treatment, just maintain hydration using Gastrolyte ® and allow the diarrhea to take its course. | 185 | 48.3% |
| Start treatment immediately with loperamide whilst drinking plenty of fluids. | 106 | 27.7% |
| Start treatment immediately with one dose of norfloxacin 800 mg plus loperamide whilst drinking plenty of fluids. | 44 | 11.55% |
| Start treatment immediately with one dose of norfloxacin 800 mg whilst drinking plenty of fluids. | 27 | 7% |
| Start treatment immediately with one dose of doxycycline 200 mg plus loperamide whilst drinking plenty of fluids | 21 | 5.5% |

them claimed that flu was prevalent among Hajj and Umrah travelers. The comprehensive replies of the CPs regarding vaccination practices during Hajj and Umrah were outlined in Table 4.

In relation to attitudes towards vaccination, the study found that 55.3% (n = 218) of CPs agreed that vaccination is a safe option for pilgrims aged 65 years and above, while 65.7% (n = 257) agreed that vaccination can help in reducing medical costs during the Hajj and Umrah seasons. Moreover, 61.1% (n = 303) of CPs recommended updating immunization

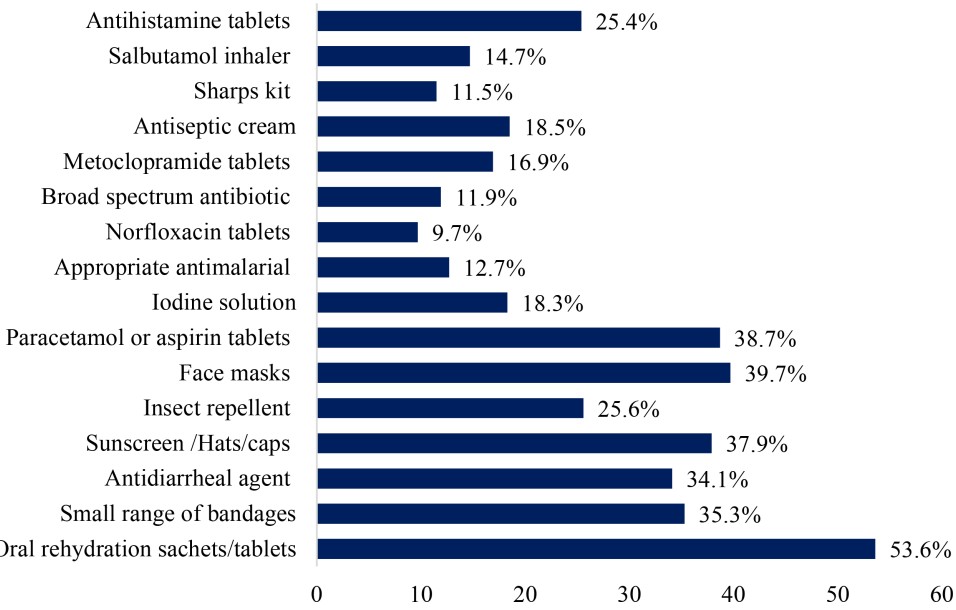

**Fig 2. CPs responses towards first aid items recommendations to pilgrims.**

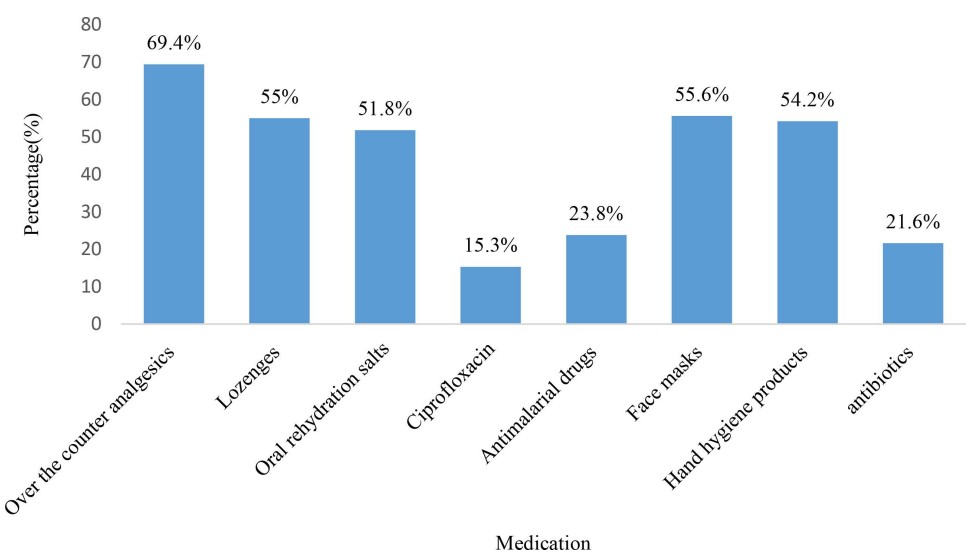

**Fig 3. CPs responses towards the most common vaccines, given before traveling to Saudi Arabia.**

**Table 4. Knowledge of CPs about the vaccination during Hajj and Umrah.**

| Questionnaires | Frequency (n) | Percentages (%) |
|---|---|---|
| **The vaccine is recommended for the protection and to prevent the spread of infectious diseases before traveling to Saudi Arabia** | | |
| Yes | 283 | 57.1% |
| No | 108 | 21.8% |
| I don't know | 105 | 21.2% |
| **Influenza is common among Hajj and Umrah travelers** | | |
| Yes | 250 | 64.8% |
| No | 57 | 14.8% |
| I don't know | 79 | 20.5% |
| **Younger adults have a higher risk of complications from influenza and other Hajj and Umrah health-related conditions.** | | |
| Yes | 133 | 34.8% |
| No | 165 | 43.2% |
| I don't know | 84 | 22% |
| **Influenza may be transmitted from person to person easily during Haj and Umrah** | | |
| Yes | 253 | 65.9% |
| No | 49 | 12.8% |
| I don't know | 82 | 21.4% |
| **The vaccine protects against influenza infections and is not obligatory to obtain Haj or Umrah visas.** | | |
| Yes | 148 | 38.4% |
| No | 129 | 33.5% |
| I don't know | 108 | 28.1% |
| Visitors arriving for Umrah or pilgrimage (Hajj) or seasonal work are required to submit a certificate of vaccination with the quadrivalent (ACYW135) vaccine against meningitis issued no more than 3 years and no less than 10 days before arrival in Saudi Arabia | | |
| Yes | 177 | 46.1% |
| No | 80 | 20.8% |
| I don't know | 127 | 33.1% |
| **The pathogen that causes influenza is divided into three types: A, B, and C** | | |
| Yes | 179 | 47.1% |
| No | 89 | 23.4% |
| I don't know | 112 | 29.5% |
| **A respiratory disease that attacks the upper and/or lower respiratory tracts was more common among Hajj and Umrah pilgrims** | | |
| Yes | 299 | 60.3% |
| No | 106 | 21.4% |
| I don't know | 91 | 18.3% |
| **Mortality due to influenza globally is low** | | |
| Yes | 129 | 33.7% |
| No | 145 | 37.9% |
| I don't know | 109 | 28.5% |
| **Children, adults, and the elderly have the same risk of infection during the Hajj and Umrah seasons** | | |
| Yes | 188 | 37.9% |
| No | 223 | 45% |
| I don't know | 85 | 17.15 |
| **Mean(Std)** | 4.739 (std = 2.49) | |
| **Median** | 5.00 | |
| **Range** | 0-9 | |

against vaccine-preventable diseases for all travelers to ensure a safe pilgrimage. The detailed responses of CPs regarding their attitudes towards vaccination are presented in Table 5. The study also found that the majority of CPs (69.4%) reported prescribing analgesics, followed by facemasks (55.6%), lozenges (55%), hand hygiene products (54.2%), oral rehydration salts (51.8%), antimalarial drugs (23.8%), and antibiotics (21.6%) for Hajj and Umrah pilgrims. The detailed prescribing pattern by CPs during Hajj and Umrah is shown in Fig 4.

Despite the mean knowledge score of 4.739(2.49) (median = 5; Range = 0-9) among CPs in this study regarding vaccination during Hajj and Umrah, the findings revealed that 56.7% of the CPs exhibited good knowledge, whereas 43.3% demonstrated poor knowledge of the subject as shown in Fig 5.

To determine the relationship between knowledge of vaccination, with community pharmacist age, gender, education level, years of experience, country of qualification, pharmacy location and Umrah related information, a multiple regression linear model was utilized in which age, gender, education level, years of experience, country of qualification, pharmacy location and Umrah related information were considered as explanatory variables and knowledge of vaccination as the dependent variable. However, there was no association between the knowledge of vaccination and the education levels of the community pharmacist, and the amount of time spent on providing Umrah related health advice, as shown in Table 6. All other variables were significantly associated with the knowledge of vaccination as shown in Table 6.

## Discussion

To the best of our knowledge, no previous research has examined the knowledge and attitudes of CPs regarding Hajj and Umrah infections in the western region of Saudi Arabia. There is limited literature available on the topic of knowledge and attitudes towards Hajj and Umrah infections, both domestically and globally. However, most of the literature has concentrated on medication use during Hajj while neglecting the subject of diseases [1,4,9]. This study has

**Table 5. Attitude of CPs regarding vaccination and other related conditions among pilgrims.**

| Variables | Frequencies (n) | Percentages (%) |
|---|---|---|
| **Vaccination is safe to be given to Hajj and Umrah perfumers aged ≥ 65 years.** | | |
| Yes | 218 | 55.3% |
| No | 99 | 25.1% |
| I don't know | 77 | 19.5% |
| **Vaccination can save medical costs during the Hajj and Umrah seasons.** | | |
| Yes | 257 | 65.7% |
| No | 58 | 14.8% |
| I don't know | 76 | 19.4% |
| **Updating immunization against vaccine-preventable diseases in all travelers is strongly recommended for safe haj and Umrah.** | | |
| Yes | 303 | 61.1% |
| No | 107 | 21.6% |
| I don't know | 86 | 17.3% |
| **Influenza vaccination for Haj and Umrah performers should be done twice a year.** | | |
| Yes | 240 | 48.4% |
| No | 148 | 29.8% |
| I don't know | 108 | 21.8% |

the potential to make a significant scientific contribution towards enhancing public health for individuals who travel from various countries to perform religious rituals at holy sites. By providing appropriate awareness and counseling from CPs, this study can serve as a valuable reference for future national and international research. Moreover, the findings can aid educational and hospital institutions in establishing appropriate continuing education efforts for practicing pharmacists to enhance their knowledge and attentiveness, especially for those working in holy cities.

According to the results of this survey, the most suggested medication for pilgrims was oral rehydration sachets (53.6%; n = 266). The next most frequently recommended items were face masks (39.7%; n = 197) and painkillers/paracetamol tablets. Approximately 39% of the CPs recommended the use of sunscreen or hats, while a limited range of bandages or dressings were suggested by 35.3% of the CPs. These findings were like those of Yezli et al. in 2020, Shakir et al. in 2006, and Stagelund et al. in 2019 [19,20,24,25]. For instance, a previous study by Yezli et al in 2020 among pilgrims visiting outpatients during the Hajj gathering reported the use of nonsteroidal anti-inflammatory drugs, paracetamol (16.61%), antihistamines (2.19%), oral rehydration salts (1.24%) and antipyretics, and antibacterial medicines [19]. Furthermore, pilgrims were prescribed an average of 2.6 medications on each consultation and polypharmacy was reported in 4.8% of the encounters [19]. Antibiotics and injections were prescribed in 46.9% and 6.5% of encounters, respectively [19]. Similarly, Shakir et al., 2006, and Stagelund et al.2019 reported that paracetamol, antibacterial, drugs acting GIT, and antihistamines were the most common drugs prescribed to pilgrims[24,25]. On the other hand, another recent study by Yezli et al in 2021 aimed to assess the Medication Handling and Storage among pilgrims during the Hajj Mass Gathering, reported that 44.4% of the pilgrims used medication and 60.2% of the pilgrims purchased from Saudi Arabia [26]. In addition, 36.6% of the medicines were prescribed by pharmacists and 78.8% of them were prescribed by physicians.

Furthermore 91.6% of the pilgrims used 1-4 medications [26]. Another study among pilgrims or visitors to Mecca reported having painkillers for Anti-diarrhea, Anti-constipation, and Antibiotic medications [27]. This indicates the seriousness of the various outbreaks caused by Bactria, viruses, and parasites, which may be the result of non-compliance with health guidelines among visitors and pilgrims [28].

However, a recent report from local news in Saudi Arabia also revealed that despite having correct vaccinations, visitors and Pilgrims who are coming to the country for Umrah or Hajj are encouraged to bring health kits as a precaution against Swine Flu [29–31]. This report on the other hand confirmed the possibility of getting various infection [29,31]. Therefore, it is advisable that visitors and pilgrims must take care of their health through adequate precautionary measures particularly for the prevention of droplet infections, by using facemasks and hand sanitizers [30–32].

According to the findings of this study, the most common illness reported during the Hajj and Umrah was diarrhea (59.5%), followed by flu and cough (58%), gastrointestinal illnesses (39.4%), food poisoning (33.6%), viral fever (24.4%), and heat stroke (24.4%). These findings were consistent with a recent study published in 2019 by Yezli et al, who showed that 9.7% of pilgrims experienced GI symptoms, while 5.1% (69/1363) of pilgrims experienced diarrhea [27]. Many other studies reported upper respiratory viral infections and bacterial respiratory infections [33]. These findings demonstrated the acquirement of enteric pathogens and parasites at mass gatherings, which may be contributing the infectious diseases [34,35]. Although it is well documented that despite the precautionary behavior and adequate sanitizing techniques followed by the pilgrims, still stomach disorders, poisoning outbreaks, and diarrhea continue to occur in the holy cities, during the mass gathering [35,36].

Even though the overall mean knowledge of CPs about vaccination during Hajj and Umrah in this study was 4.739(2.49) (median = 5; Range = 0-9). However, the study found that 56.7% of the CPs had good knowledge, while 43.3% had poor knowledge of vaccination during Hajj and Umrah. The greatest knowledge score was reported on how common influenza illness is during Umrah and Hajj (64.8%), followed by person-to-person transmission being the main mode of transmission during Haj and Umrah (65.9%). Although the knowledge of levels may differ from one study to another which may be influenced by several factors including the study method, respondents, and demographics of the subjects, it is evidenced that practicing pharmacists would be found to have good knowledge and then student pharmacists. In

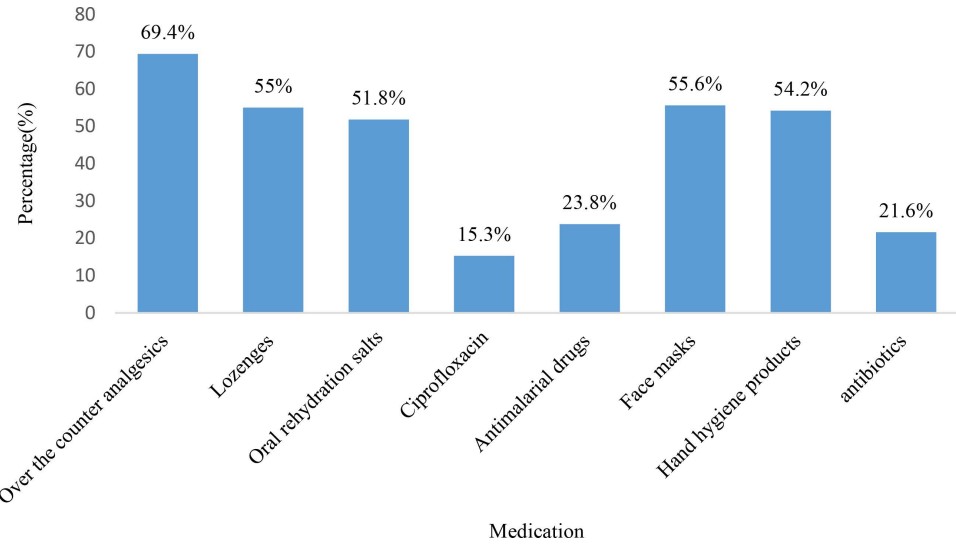

**Fig 4. Commonly prescribed medication for Haj and Umrah-related health conditions.**

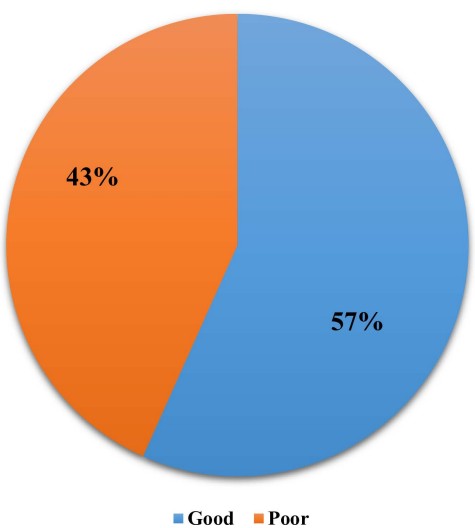

**Fig 5. Knowledge levels of CPs regarding the vaccination.**

addition, various educational interventions should be recommended for the practicing pharmacist to get up-to-date information on the prevention and management of various infectious diseases, which may help in controlling and adequate management of the diseases among pilgrims and patients.

Vaccinations were given to international pilgrims to prevent influenza and other diseases [33]. Influenza is seen as a disease that may be avoided with effective vaccination, hence it is recommended for all individuals above the age of six months [35,37]. Elderly individuals, as well as those with chronic medical conditions, were more likely to develop a serious case of influenza. Individuals with chronic disease conditions and the elderly should get the seasonal influenza vaccine each year before traveling [2,35]. Our findings revealed that although the majority of CPs were aware of the essential vaccines for Hajj and Umrah before coming to Saudi Arabia, CPs' knowledge of various illnesses and vaccinations might be improved. As of now, no study has been published to assess CPs' knowledge, and attitudes toward various infections, management, and vaccination during Haj and Umrah, thus, we cannot make a direct comparison with other studies however, a previous study among health care professionals in Saudi Arabia reveals a positive attitude toward vaccination, as the majority believes vaccination is effective in preventing and shortening the duration of infection and remission [38].

The current study has some limitations that should be taken into account. Firstly, the research was confined to only one region in the western province of Saudi Arabia. Therefore, the generalizability of these findings to other regions may be limited. Additionally, the use of self-reported questionnaires may introduce social desirability bias in the responses provided, therefore future studies with other data collection methods are required to provide more accurate responses. Despite these limitations, the study has several strengths. Firstly, it highlights the crucial role of CPs in providing adequate treatment and management for various illnesses during mass gatherings in holy cities. Furthermore, there is a lack of research focusing on CPs and their involvement in vaccinations and infectious diseases in Saudi Arabia. By focusing on CPs, this study provides valuable insights into how their knowledge could improve health outcomes for pilgrims and facilitate better treatment. Finally, future research should aim to

**Table 6. Results of knowledge of vaccination with Demographic characters of the community pharmacist using regression analysis.**

| Variables | Unstandardized Coefficients | | Standardized Coefficients | t | Sig. | 95.0% Confidence Interval for B | |
|---|---|---|---|---|---|---|---|
| | B | Std. Error | Beta | | | Lower Bound | Upper Bound |
| (Constant) | 2.500 | .189 | | 13.197 | .000 | 2.127 | 2.872 |
| Gender | -.230 | .056 | -.194 | -4.084 | .000 | -.340 | -.119 |
| Age (Years) | .053 | .027 | .112 | 2.000 | .046 | .001 | .106 |
| Educational level | -.056 | .031 | -.082 | -1.793 | .074 | -.117 | .005 |
| Years of experience in community pharmacy | -.057 | .028 | -.115 | -2.070 | .039 | -.112 | -.003 |
| Country of qualification | -.058 | .019 | -.145 | -3.035 | .003 | -.095 | -.020 |
| Pharmacy Location | .093 | .045 | .092 | 2.071 | .039 | .005 | .181 |
| Do you provide Hajj and Umrah health-related conditions advice or services to the patients? | -.204 | .030 | -.323 | -6.731 | .000 | -.264 | -.144 |
| Have You Seen or received a pilgrim to the pharmacy about Complaining of infection/ problem? | -.137 | .045 | -.141 | -3.048 | .002 | -.226 | -.049 |
| How much time or days you spend on providing Hajj and Umrah health-related conditions advice pilgrims. | .013 | .011 | .053 | 1.112 | .267 | -.010 | .035 |

conduct comprehensive surveys that allow pharmacists from all regions of Saudi Arabia with varying levels of experience to participate.

## Conclusion

In summary, community pharmacists (CPs) in the Mecca region were found to have good knowledge, with more than half of them having positive attitudes about vaccination for Hajj and Umrah-related health conditions. Most CPs agreed that a vaccine is recommended before traveling to Saudi Arabia for protection and to prevent the spread of infectious diseases. The most common infectious diseases in the holy cities were diarrhea, followed by flu, cough, gastrointestinal illnesses, food poisoning, viral fever, and heat stroke. Furthermore, CPs' attitudes toward vaccination need to be improved. There is a need to encourage pharmacists to engage in planned and focused ongoing education programs on the management of various diseases at holy cities during mass gatherings. Additionally, it is recommended that individuals and visitors be more vigilant towards the possibility of infections through awareness and education about safety and precautionary measures with the help of healthcare providers such as pharmacists or physicians to protect against the various infectious diseases that may be present during mass gatherings.

## Supporting information

**S1 File. Hajj & Umrah.**
(XLSX)

## Author contributions

**Conceptualization:** Ahmed M. Ashour.

**Formal analysis:** Fahad S. Alshehri.

**Resources:** Maan H. Harbi.

**Visualization:** Nasser M. Alorfi.

**Writing – original draft:** Nasser M. Alorfi.

**Writing – review & editing:** Nasser M. Alorfi.

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
