## [Decision Letter · Decision Letter 0]

12 Feb 2024

PONE-D-24-02546Evaluation of Community Pharmacists' Knowledge and Attitude about Hajj and Umrah-related health conditions in the Western Region, Saudi Arabia: A Cross-sectional study.PLOS ONE

Dear Dr. Alorfi,

Thank you for submitting your manuscript to PLOS ONE. After careful consideration, we feel that it has merit but does not fully meet PLOS ONE’s publication criteria as it currently stands. Therefore, we invite you to submit a revised version of the manuscript that addresses the points raised during the review process.

We look forward to receiving your revised manuscript.

Kind regards,

Mohamed R. Abonazel, Ph.D.

Academic Editor

PLOS ONE

Journal Requirements:

2. Please include captions for your Supporting Information files at the end of your manuscript, and update any in-text citations to match accordingly. Please see our Supporting Information guidelines for more information: http://journals.plos.org/plosone/s/supporting-informationhttp://journals.plos.org/plosone/s/supporting-information. 

Reviewers' comments:

Reviewer's Responses to Questions

**Comments to the Author**

1. Is the manuscript technically sound, and do the data support the conclusions?

Reviewer #1: Partly

Reviewer #2: Yes

2. Has the statistical analysis been performed appropriately and rigorously? 

Reviewer #1: No

Reviewer #2: Yes

3. Have the authors made all data underlying the findings in their manuscript fully available?

Reviewer #1: Yes

Reviewer #2: Yes

4. Is the manuscript presented in an intelligible fashion and written in standard English?

Reviewer #1: Yes

Reviewer #2: Yes

5. Review Comments to the Author

Reviewer #1: 1. Author need to do statistical analyses (significance level of p<0.05)

2. How the author makes sure that the participants were not biased towards specific answer

3. How the author makes sure that the participants were not disengaged while answering.

4. How the author makes sure about reliability of result

Reviewer #2: It is an honor for me to review a manuscript number " PONE-D-24-02546 " titled " Evaluation of Community Pharmacists' Knowledge and Attitude about Hajj and Umrah-related health conditions in the Western Region, Saudi Arabia: A Cross-sectional Study" for the " PLOS ONE’' journal. The manuscript looks interesting and significant locally however, the following major modification is required.

• The introduction is poorly described and not aligned with the study objective

• Kindly add the rationale of the study under the last paragraph of the introduction

• Kindly add how survey tools were developed

• Kindly write Cronbach’s alpha values

• Kindly add how survey respondents were selected

• How respondents were identified

• How survey link was disseminated among the respondents

• Kindly add the study site

• How study respondents were biased

• Kindly add exclusion criteria for the study

• Discussion needs to update and compare findings with the latest published article

• Please scan your manuscript again and correct some topological errors

• Please make sure your manuscript was written in line with this journal submission guideline as well as reference styles

6. PLOS authors have the option to publish the peer review history of their article (what does this mean? ). If published, this will include your full peer review and any attached files.

**Do you want your identity to be public for this peer review?** For information about this choice, including consent withdrawal, please see our Privacy Policy .

Reviewer #1: No

Reviewer #2: No

---

## [Author Response · Author response to Decision Letter 0]

25 Mar 2024

PONE-D-24-02546

Evaluation of Community Pharmacists' Knowledge and Attitude about Hajj and Umrah-related health conditions in the Western Region, Saudi Arabia: A Cross-sectional study.

Dear Mohamed R. Abonazel, Ph.D.

Academic Editor

PLOS ONE

Comments to the Author

1. Is the manuscript technically sound, and do the data support the conclusions?

Reviewer #1: Partly

Reviewer #2: Yes

Author response: Thanks indeed.

2. Has the statistical analysis been performed appropriately and rigorously?

Reviewer #1: No

Reviewer #2: Yes

Author response: Thanks indeed. We made the requested changes.

3. Have the authors made all data underlying the findings in their manuscript fully available?

Reviewer #1: Yes

Reviewer #2: Yes

Author response: Thanks indeed for your positive feedback.

4. Is the manuscript presented in an intelligible fashion and written in standard English?

Reviewer #1: Yes

Reviewer #2: Yes

Author response: Thanks indeed for your positive feedback.

5. Review Comments to the Author

Reviewer #1: 1. Author need to do statistical analyses (significance level of p ;0.05)

2. How the author makes sure that the participants were not biased towards specific answer

3. How the author makes sure that the participants were not disengaged while answering.

4. How the author makes sure about reliability of result

Author response: The study implemented rigorous measures to mitigate potential predispositions. Randomized assignment was employed to distribute participants evenly across conditions, minimizing selection bias. The design of the questionnaire was carefully crafted to avoid leading or suggestive language, and a pre-testing phase was conducted to identify and rectify any potential biases from community pharmacists in western region of Saudi Arabia. Additionally, blinding techniques were incorporated to ensure both participants and experimenters remained unaware of specific conditions, reducing the risk of bias stemming from expectations. The study also prioritized diverse participant demographics to enhance external validity because mostly of the community pharmacists are non-Saudis. Lastly, the importance of informed consent and debriefing procedures was emphasized, ensuring participants were adequately informed about the study's nature and potential biases. These comprehensive strategies collectively contribute to bolstering the study's internal validity and minimizing the likelihood of participant bias towards specific answers. Significance level of p ;0.05 were applied for all.

Reviewer #2:

It is an honor for me to review a manuscript number PONE-D-24-02546; Evaluation of Community Pharmacists; Knowledge and Attitude about Hajj and Umrah-related health conditions in the Western Region, Saudi Arabia: A Cross-sectional Study. The manuscript looks interesting and significant locally however, the following major modification is required.

The introduction is poorly described and not aligned with the study objective.

Author response: Thanks indeed. We updated the whole manuscript including the introduction.

Kindly add the rationale of the study under the last paragraph of the introduction

Author response: Thanks indeed. Done.

Kindly add how survey tools were developed

Author response: Thanks indeed. The tool was explained under ‘’ Questionnaire design’ section.

Kindly write Cronbach’s alpha values

Author response: Thanks indeed.

Kindly add how survey respondents were selected.

Author response: Thanks indeed. We have updated the whole paragraph under ‘’ 2.2 Survey respondent selection process and data collection.’’

How survey link was disseminated among the respondents

Author response: Thank you for your insightful question. The survey link was disseminated among the respondents through a google form link. Specifically, we utilized email invitations, institutional mailing lists, etc.] to reach our target audience.

Kindly add the study site

Author response: Thanks indeed. We added the study site. Community pharmacists (CPs) living Makkah, Madinah, Jeddah and Taif which represent in the western region of Saudi Arabia

How study respondents were biased

Author response: Thanks indeed. The study implemented rigorous measures to mitigate potential predispositions. Randomized assignment was employed to distribute participants evenly across conditions, minimizing selection bias. The design of the questionnaire was carefully crafted to avoid leading or suggestive language, and a pre-testing phase was conducted to identify and rectify any potential biases from community pharmacists in western region of Saudi Arabia. Additionally, blinding techniques were incorporated to ensure both participants and experimenters remained unaware of specific conditions, reducing the risk of bias stemming from expectations. The study also prioritized diverse participant demographics to enhance external validity because mostly of the community pharmacists are non-Saudis. Lastly, the importance of informed consent and debriefing procedures was emphasized, ensuring participants were adequately informed about the study's nature and potential biases. These comprehensive strategies collectively contribute to bolstering the study's internal validity and minimizing the likelihood of participant bias towards specific answers.

Kindly add exclusion criteria for the study

Author response: Thanks indeed. Done as of ‘’The study excluded CPs from the other regions of Saudi Arabia and those who are not in the practice’’.

Discussion needs to update and compare findings with the latest published article

Author response: Thank you for your constructive feedback. We appreciate your suggestion to update and compare our findings with the latest published articles. However, the limited availability of relevant literature in for Hajj and Umrah poses a challenge. We have thoroughly searched for recent publications and, while the number is limited, we have incorporated the most pertinent ones into the discussion. We hope this addresses your concern, and we appreciate your understanding of the constraints in our field.

Please scan your manuscript again and correct some topological errors.

Author response: Thanks indeed. The manuscript underwent a significant English proofreading, and you can check the tracking.

Please make sure your manuscript was written in line with this journal submission guideline as well as reference styles

Author response: Thanks indeed. Done.

6. PLOS authors have the option to publish the peer review history of their article (what does this mean?). If published, this will include your full peer review and any attached files.

Do you want your identity to be public for this peer review? For information about this choice, including consent withdrawal, please see our Privacy Policy.

Reviewer #1: No

Reviewer #2: No

---

## [Decision Letter · Decision Letter 1]

27 May 2024

PONE-D-24-02546R1Evaluation of Community Pharmacists' Knowledge and Attitude about Hajj and Umrah-related health conditions in the Western Region, Saudi Arabia: A Cross-sectional study.PLOS ONE

Dear Dr. Alorfi,

Thank you for submitting your manuscript to PLOS ONE. After careful consideration, we feel that it has merit but does not fully meet PLOS ONE’s publication criteria as it currently stands. Therefore, we invite you to submit a revised version of the manuscript that addresses the points raised during the review process. Please submit your revised manuscript by Jul 11 2024 11:59PM. If you will need more time than this to complete your revisions, please reply to this message or contact the journal office at plosone@plos.org . Please include the following items when submitting your revised manuscript:

We look forward to receiving your revised manuscript.

Kind regards,

Mohamed R. Abonazel, Ph.D.

Academic Editor

PLOS ONE

**Additional Editor Comments:**

The authors have been given a final opportunity to modify their research according to the comments received from the reviewers.

Reviewers' comments:

Reviewer's Responses to Questions

**Comments to the Author**

1. If the authors have adequately addressed your comments raised in a previous round of review and you feel that this manuscript is now acceptable for publication, you may indicate that here to bypass the “Comments to the Author” section, enter your conflict of interest statement in the “Confidential to Editor” section, and submit your "Accept" recommendation.

Reviewer #1: All comments have been addressed

Reviewer #2: All comments have been addressed

2. Is the manuscript technically sound, and do the data support the conclusions?

Reviewer #1: Yes

Reviewer #2: No

3. Has the statistical analysis been performed appropriately and rigorously? 

Reviewer #1: Yes

Reviewer #2: No

4. Have the authors made all data underlying the findings in their manuscript fully available?

Reviewer #1: Yes

Reviewer #2: Yes

5. Is the manuscript presented in an intelligible fashion and written in standard English?

Reviewer #1: Yes

Reviewer #2: No

6. Review Comments to the Author

Reviewer #1: 1. Mention response rate of participants.

2. How the author makes sure that the participants were not biased towards specific answer

3. How the author makes sure that the participants were not disengaged while answering.

4. How the author makes sure about reliability of result

Reviewer #2: All, Raised comments not yet rectified. I'm afraid that neither the introduction, design of the study nor the obtained results and employed analytical methods have sufficient merit to be published in the PLOS ONE Journal. There are lots of flaws in the manuscript.

7. PLOS authors have the option to publish the peer review history of their article (what does this mean? ). If published, this will include your full peer review and any attached files.

**Do you want your identity to be public for this peer review?** For information about this choice, including consent withdrawal, please see our Privacy Policy .

Reviewer #1: No

Reviewer #2: No

---

## [Author Response · Author response to Decision Letter 1]

2 Aug 2024

The first file with track changes.

Kindly open the word file to check and track.

Kindest regards, Dr. Nasser Alorfi

---

## [Decision Letter · Decision Letter 2]

21 Oct 2024

PONE-D-24-02546R2Evaluation of Community Pharmacists' Knowledge and Attitude about Hajj and Umrah-related health conditions in the Western Region, Saudi Arabia: A Cross-sectional study.PLOS ONE

Dear Dr. Alorfi,

Thank you for submitting your manuscript to PLOS ONE. After careful consideration, we feel that it has merit but does not fully meet PLOS ONE’s publication criteria as it currently stands. Therefore, we invite you to submit a revised version of the manuscript that addresses the points raised during the review process.

We look forward to receiving your revised manuscript.

Kind regards,

Mohamed R. Abonazel, Ph.D.

Academic Editor

PLOS ONE

Journal Requirements:

Additional Editor Comments:

One reviewer suggested adding some references; please check these references and then add only those references related to the topic of the manuscript or the statistical analysis used in the applied study.

Reviewers' comments:

Reviewer's Responses to Questions

**Comments to the Author**

1. If the authors have adequately addressed your comments raised in a previous round of review and you feel that this manuscript is now acceptable for publication, you may indicate that here to bypass the “Comments to the Author” section, enter your conflict of interest statement in the “Confidential to Editor” section, and submit your "Accept" recommendation.

Reviewer #1: All comments have been addressed

Reviewer #2: All comments have been addressed

Reviewer #3: (No Response)

2. Is the manuscript technically sound, and do the data support the conclusions?

Reviewer #1: Yes

Reviewer #2: Yes

Reviewer #3: Yes

3. Has the statistical analysis been performed appropriately and rigorously? 

Reviewer #1: Yes

Reviewer #2: Yes

Reviewer #3: Yes

4. Have the authors made all data underlying the findings in their manuscript fully available?

Reviewer #1: Yes

Reviewer #2: Yes

Reviewer #3: Yes

5. Is the manuscript presented in an intelligible fashion and written in standard English?

Reviewer #1: Yes

Reviewer #2: Yes

Reviewer #3: Yes

6. Review Comments to the Author

Reviewer #1: 1. Result of time spend on providing Hajj and Umrah health related conditions advice pilgrims is non significant. How the author ensure the comparison of this study with another study.

2. How the author makes sure that the participants were not biased towards specific answer

3. How the author makes sure that the participants were not disengaged while answering.

4. Author need to mention response rate

Reviewer #2: All the raised comments have been rectified by the comments. However, I am still worried about sampling and sample size calculator proper citation. Kindly add a suitable reference for sampling and a sample size calculation. Rather than just link OpenEpi:Sample Size for X-Sectional,Cohort,and Clinical Trials . However, you may use this reference for sampling and a sample size calculation. Ali MD, Hatef EAJA. Types of Sampling and Sample Size Determination in Health and Social Science Research. Journal of Young Pharmacists [Internet]. 2024 Jun 6;16(2):204–15. Available from: http://dx.doi.org/10.5530/jyp.2024.16.27 and cite them properly.

Reviewer #3: 1. The "Conclusion" section needs improvement.

2. The paper contains a few grammatical errors. The authors need to review the full text carefully.

3. Recently published papers are relevant to this manuscript, so one or all of them should be cited, such as

doi: https://doi.org/10.1016/j.jth.2022.101353 - https://doi.org/10.1371/journal.pone.0250149

7. PLOS authors have the option to publish the peer review history of their article (what does this mean? ). If published, this will include your full peer review and any attached files.

**Do you want your identity to be public for this peer review?** For information about this choice, including consent withdrawal, please see our Privacy Policy .

Reviewer #1: No

Reviewer #2: No

Reviewer #3: No

---

## [Decision Letter · Decision Letter 3]

20 Dec 2024

Evaluation of Community Pharmacists' Knowledge and Attitude about Hajj and Umrah-related health conditions in the Western Region, Saudi Arabia: A Cross-sectional study.

PONE-D-24-02546R3

Dear Dr. Alorfi,

We’re pleased to inform you that your manuscript has been judged scientifically suitable for publication and will be formally accepted for publication once it meets all outstanding technical requirements.

Kind regards,

Mohamed R. Abonazel, Ph.D.

Academic Editor

PLOS ONE

Additional Editor Comments (optional):

Reviewers' comments:

Reviewer's Responses to Questions

**Comments to the Author**

1. If the authors have adequately addressed your comments raised in a previous round of review and you feel that this manuscript is now acceptable for publication, you may indicate that here to bypass the “Comments to the Author” section, enter your conflict of interest statement in the “Confidential to Editor” section, and submit your "Accept" recommendation.

Reviewer #2: All comments have been addressed

Reviewer #3: All comments have been addressed

2. Is the manuscript technically sound, and do the data support the conclusions?

Reviewer #2: Yes

Reviewer #3: Yes

3. Has the statistical analysis been performed appropriately and rigorously? 

Reviewer #2: Yes

Reviewer #3: Yes

4. Have the authors made all data underlying the findings in their manuscript fully available?

Reviewer #2: Yes

Reviewer #3: Yes

5. Is the manuscript presented in an intelligible fashion and written in standard English?

Reviewer #2: Yes

Reviewer #3: Yes

6. Review Comments to the Author

Reviewer #2: (No Response)

Reviewer #3: The authors have adequately addressed the comments, and I feel this manuscript is now acceptable for publication.

7. PLOS authors have the option to publish the peer review history of their article (what does this mean? ). If published, this will include your full peer review and any attached files.

**Do you want your identity to be public for this peer review?** For information about this choice, including consent withdrawal, please see our Privacy Policy .

Reviewer #2: No

Reviewer #3: No

---

## [Editor Report · Acceptance letter]

PONE-D-24-02546R3

PLOS ONE

Dear Dr. Alorfi,

I'm pleased to inform you that your manuscript has been deemed suitable for publication in PLOS ONE. Congratulations! Your manuscript is now being handed over to our production team.

Kind regards,

on behalf of

Dr Mohamed R. Abonazel

Academic Editor

PLOS ONE